# Designing intervention prototypes to improve infant and young child nutrition in Peru: a participatory design study protocol

Emily Rousham [ID],[1] Rossina G Pareja,[2] Hilary M Creed-Kanashiro,[2] Rosario Bartolini,[2] Rebecca Pradeilles,[1] Deysi Ortega-Roman,[3] Michelle Holdsworth,[4] Paula Griffiths,[1] Nervo Verdezoto[3]

ER and RGP contributed equally.

## ABSTRACT

**Introduction** Nutrition during the complementary feeding period (6–23 months) is critical to ensure optimal growth and reduce the risk of diet-related disease across the life course. Strategies to reduce multiple forms of malnutrition (stunting, overweight/obesity and anaemia) in infants and young children (IYC) are a key priority in low-income and middle-income countries, including Peru. This study aims to co-design and develop prototypes for interventions to address the multiple forms of malnutrition in IYC in urban Peru, using a participatory design approach.

**Methods and analysis** The study will be based within peri-urban communities in two areas of Peru (Lima and Huánuco city). Following the identification of key nutritional challenges for IYC aged 6–23 months through formative research (phase I), we will conduct a series of workshops bringing together healthcare professionals from government health centres and caregivers of IYC aged 6–23 months. Workshops (on idea generation; creating future scenarios; storyboarding and early implementation and feedback) will take place in parallel in the two study areas. Through these workshops, we will engage with community participants to explore, experiment, co-design and iteratively validate new design ideas to address the challenges around IYC complementary feeding from phase I. Workshop outputs and transcripts will be analysed qualitatively using affinity diagramming and thematic analyses. The intervention prototypes will be evaluated qualitatively and piloted with the participating communities.

**Ethics and dissemination** Ethical approval for this study was obtained from the Ethical Review Committee of the Instituto de Investigación Nutricional (IIN) Peru (388-2019/CIEI-IIN), Loughborough University (C19-87) and confirmed by Cardiff University. Findings of the participatory design process will be disseminated through a deliberative workshop in Lima, Peru with national and regional government stakeholders, as well as participants and researchers involved in the design process. Further dissemination will take place through policy briefs, conferences and academic publications.

## STRENGTHS AND LIMITATIONS OF THIS STUDY

⇒ Participatory co-design is an important way of incorporating the voice of the community in the design and development of health interventions.
⇒ This participatory study will bring together healthcare providers and health service users in collaborative co-design workshops to promote active community involvement in infant and young child nutrition, and mutual learning for all participants including the researchers.
⇒ The co-designed prototypes will be tailored to the nutritional challenges for infants and young children in peri-urban Peru.
⇒ The experiences and ideas expressed by participants may be influenced by the impact of the COVID-19 pandemic on health services and access to healthcare in the previous 2 years.

## INTRODUCTION

Nutrition during the complementary feeding period (6–23 months) is critical to ensure optimal growth and reduce the risk of diet-related diseases across the life course. In low-resource settings, the prevention of undernutrition in the form of stunting, wasting or micronutrient deficiencies has been the traditional focus of complementary feeding guidance.[1–3] However, as populations go through dietary and nutrition transitions, infant and young child (IYC) diets are changing with increased consumption of sugar, fats and oils, refined grains and highly processed foods, bringing the additional risk of excess energy intake leading to overweight and obesity.[4–7] This in turn leads to increased risk of diet-related diseases in later life. Strategies are now needed to address coexisting forms of malnutrition in infancy and childhood including stunting, micronutrient deficiencies and overweight/obesity.[7–9]

[1]Loughborough University, Loughborough, UK
[2]Instituto de Investigación Nutricional, Lima, Peru
[3]Cardiff University, Cardiff, UK
[4]NUTRIPASS, IRD, Marseille, France

**Correspondence to**
Dr Emily Rousham;
e.k.rousham@lboro.ac.uk

Peru faces the public health challenge of multiple burdens of malnutrition with a high prevalence of iron-deficiency anaemia, stunting (low height-for-age) and a rapidly increasing prevalence of overweight and obesity.[10] Peru has successfully reduced the rate of stunting in children under 5 years from 23.2% in 2010 to 12.1% in 2020.[10] This success has been attributed to antipoverty policies, multisectoral government working and implementation of cross-cutting interventions in the poorest areas.[11] In contrast, anaemia prevalence has changed little among children aged 6–36 months from 43.5% in 2015 to 40.0% in 2020[10][12], despite this being a national priority. Since 2017, government strategies to address anaemia have focused on national distribution of multimicronutrient powders and iron supplementation for IYC through health services and promotion of iron-rich foods.[13]

Additional concerns focus on the increasing prevalence of overweight and obesity in Peru, affecting 37.4% of children aged 5–9 years.[14] Cohort studies in Peru have shown rapid weight gain in infancy is associated with overweight and obesity at 8 years.[15] Feeding practices in infancy or early childhood also shape future dietary behaviours, such that unhealthy complementary feeding practices may predispose to unhealthy diets in later life.[6]

Although previous research has examined the influence of sociocultural contexts on complementary feeding practices in relation to undernutrition,[16] less attention has been given to exploring these influences on excess energy intake and overweight during early life in Peru. Similarly, there are no integrated policies or interventions to prevent overweight alongside iron and micronutrient deficiencies in IYC.

Sociotechnical approaches to healthcare technology design and implementation, such as human-centred design, co-design and community-based participatory research, have led to an increasing involvement of multiple stakeholders in the design process.[17–19] Such methods can help multiple stakeholders, including community members, to express their thoughts, share ideas, experiment and actively participate[20] throughout the design process.[21] However, most of these sociotechnical initiatives have taken place in the Global North[22 23] highlighting the need for the development of new methods and tools to engage with Global South communities. While some previous studies have applied participatory co-design methods in Latin American or Peruvian contexts,[24–28] few have involved technology in healthcare settings.[29]

The aim of this study is to co-design and develop prototypes for interventions to address the multiple forms of malnutrition coexisting in IYC in Peru, using a participatory design approach. This co-design follows on from formative research in the communities and health services that used qualitative and quantitative data as well as policy mapping to assess the extent of malnutrition in peri-urban communities, underlying contributors to malnutrition and the key nutritional challenges identified by researchers and community participants.

## METHODS AND ANALYSIS
### Study background and phase I
The PERUSANO (Perú - Estrategias y opoRtUnidad: Sin Anemia Ni Obesidad) study is an interdisciplinary project which aims to address multiple forms of malnutrition among IYC aged 6–23 months in peri-urban communities in Peru. This protocol follows on from the first phase of research in the PERUSANO project. The formative research (phase I) aimed to identify the prevalence of all

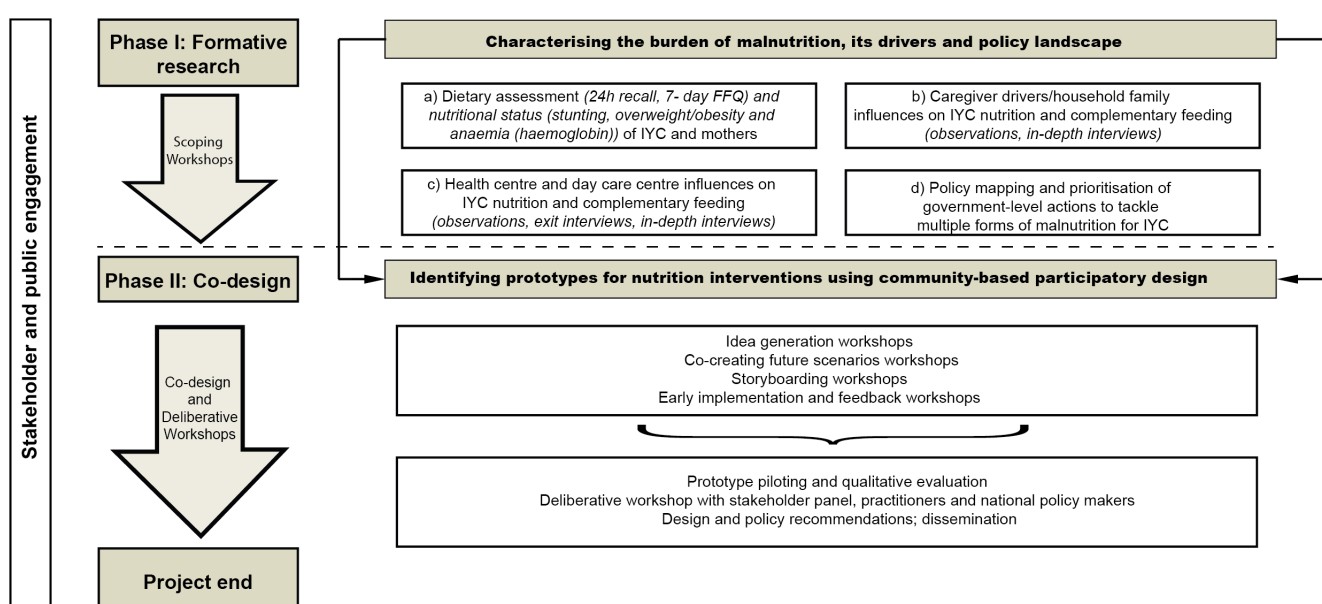

**Figure 1** Overview of the PERUSANO study; phase I summarises the formative research objectives and phase II outlines the protocol for the participatory co-design. FFQ, food frequency questionnaire; IYC, infants and young children.

forms of malnutrition as well as the challenges and facilitators of healthy IYC feeding practices. This informed the second phase of the project that will take a co-design approach with community participants (healthcare professionals, mothers and family caregivers, and other community actors) to scope opportunities to address the identified IYC feeding challenges. The wider PERUSANO project team includes a cross-cultural and multidisciplinary group of researchers (7 from Peru and 10 from UK/France) with backgrounds in public health, nutrition, social sciences and human-computer interaction (HCI). Phase I objectives were achieved through four work packages: (a) cross-sectional, quantitative assessment of dietary intake (via a 7-day food frequency questionnaire (FFQ) for mothers and a quantitative 24-hour dietary recall for IYC) and nutritional status (weight, length/height and haemoglobin concentration) of IYC and mothers alongside a household sociodemographic survey (n=244); (b) household in-depth qualitative studies (semi-structured interviews, observations) of caregiver drivers of IYC feeding (eg, feeding style, food attributes), (c) interviews and observations of healthcare professionals responsible for nutritional counselling and well-baby clinics and (d) policy mapping and prioritisation of government-level actions that exist to tackle multiple forms of malnutrition for IYC with 16 national experts. An overview of the study design with phase I work packages and phase II components is shown in figure 1.

Phase I started in December 2019 and was interrupted by the COVID-19 pandemic in March 2020. COVID-19 restrictions in Peru prevented face-to-face workshops or field work taking place from March 2020 to May 2022. During this time, we analysed data from phase I and continued meetings with researchers and stakeholders via online platforms. An 11-month no-cost extension to the project was granted by funders due to the impact of COVID-19, which enabled face-to-face data collection to commence again in June 2022.

### End of phase I (predesign): identification of key infant and young child nutritional challenges for participatory co-design

In July 2021, we conducted an online scoping workshop with 19 members of the interdisciplinary team, where researchers presented the nuanced insights and major challenges uncovered during phase I. Before the scoping workshop, the challenges identified from the four formative work packages (a–d above) and supporting literature were visualised using an online collaborative whiteboard platform (Miro) and then reorganised and refined during two sessions held among the research team with the use of affinity diagramming.[30 31] Affinity diagramming is a collaborative sense-making technique that supports a bottom-up approach for qualitative data analysis where data are written on separate sticky notes and categorised/clustered around insights based on their affinity.[32–34] At this stage, there were 27 challenges which were grouped into three overarching themes: (1) the local healthcare system (10 challenges), (2) the national healthcare

> **Box 1** Top 10 opportunities for intervention in infant and young child (IYC) nutrition identified and prioritised through formative research and scoping workshops to take forward to the participatory design phase
>
> **Opportunity statements identified and prioritised by the research team\*.**
> 1. Reduce the consumption of unhealthy foods and beverages among mothers and IYC (6–23 months) to improve the overall quality of diet/feeding practices.
> 2. Develop strategies to reduce inequalities to protect vulnerable populations in relation to the double burden of malnutrition.
> 3. Improve emotional and psychological well-being for mothers to reduce its impact on IYC nutrition.
> 4. Increase/Enhance maternal nutrition knowledge on appropriate feeding practices (eg, iron supplementation, complementary feeding, breast feeding).
> 5. Help to implement governmental policies for early childhood education services to provide and promote healthy food choices.
> 6. Increase adherence to iron supplementation for IYC.
> 7. Enhance the uptake, access and intercultural approaches to improve maternal and child care services.
> 8. Enhance counselling skills for healthcare professionals.
> 9. Improve nutrition counselling during pregnancy and from birth onwards.
> 10. Address the challenges in coordinating the implementation and evaluation of policy across different levels of governmental institutions in relation to the double burden of malnutrition.
>
> \*The full list of opportunities which were voted on to prioritise the 10 opportunities is provided in online supplemental table 1.

system (10 challenges) and (3) maternal and IYC nutrition and feeding practices (9 challenges). Three challenges featured under more than one of the overarching themes, as shown in online supplemental table 1.

To help define specific issues and narrow down the scope for intervention, we transformed the challenges into opportunities and thematically grouped them, resulting in 19 opportunities (online supplemental table 1). To assist with prioritising, we asked all research team participants to vote independently for their top 4 (out of 19) most important opportunities from their own perspectives during the scoping workshop. Miro automatically counted the votes and displayed the total votes and the final ranking. Two members who could not attend the workshop voted remotely and these were subsequently added to the total. The votes were used to select the top 10 opportunities (box 1). We then allocated one of the top-ranking opportunities using the "How Might We" technique[35–37] to each of the four breakout rooms during the scoping workshop. Each breakout room had a mix of interdisciplinary researchers who then reflected on one of the top particular challenges/opportunities and identified potential strategies to address them.

After the scoping workshop, we undertook a final round of analysis using affinity diagramming to select opportunities to take forward for exploration and scoping with the community participants. Out of the top 10 most voted

opportunities and their associated challenges, a final group of 4 major challenges were selected for phase II:

► High consumption of unhealthy foods, sugar-sweetened beverages and savoury snacks (fried, salty foods, sweet products) in mothers and IYC.
► Low prevalence of (or difficulties with) iron supplementation in IYC.
► Issues with nutritional counselling and maternal well-being.
► No way of tracking the double burden of malnutrition.

### Phase II: identification of prototypes for nutrition interventions using community-based participatory design

#### Patient and public involvement

The study has involved local, regional and national stakeholders since the early stages of developing phase I protocols. Consultation meetings took place with national and regional stakeholders at the inception of the project, and we have continued with regular consultations and presentations of the phase I findings to municipal (regional) government authorities and national government officials in ministries responsible for health and social development. In Huánuco, in addition to the regional authorities, we are also working with the '*Mesa*' (Mesa de Concertación de Lucha contra la Pobreza) which is the coordinating body for civic society organisations.

Through national and local consultations prior to phase II, we have been able to develop ways of making the participatory co-design accessible and inclusive. For example, workshops will be scheduled at a time of day that is preferred by caregivers to accommodate other commitments (typically 14:30–16:30 hours or 15.00–17.00 hours). Assistance with transport costs will be provided as that was identified as a barrier to participation. Community health agents will liaise with participants before and after workshops to give reminders of date, time and location as they are trusted within the communities. Community health agents will also assist with care of IYC and older siblings during workshops to allow caregivers full participation.

### Study design

Participatory design activities will be led by researchers in HCI and design (NV and DO-R) with input from the multidisciplinary research team leads from phase I with expertise in nutrition (RGP, HMC-K, MH, ER, RP, PG), nutrition policy (MH, RGP), anthropology and social sciences (RB, HMC-K, PG, ER, RGP). The PERUSANO team has previous experience ranging from >20 years of research in low-income and middle-income community settings to mid-career and early career researchers, including doctoral researchers in Peru and in the UK. The team includes researchers fluent in Spanish, English and both languages (NV, HMC-K, RGP, DO-R). The participatory design phase started in June 2022 and will be completed in March 2023.

### Methodological framework

Participatory design is a research approach for the development of technology for real-world problems with the main principle to actively involve users in the design decisions of the systems designed, initially rooted within the workplace in Scandinavia,[38 39] and later to support everyday life.[22 40] Participatory design has shown its potential as a research approach to effectively bridge the gap between computer technologies and health-related interventional research,[41] fostering empathy and mutual learning among different stakeholders.[42] Participatory design has been used in many different domains including public health and healthcare in clinical and non-clinical settings using a range of different methods to support healthcare technology design.[23]

### Study settings

The project focuses on the rapidly urbanising population of Peru constituting 78% of the national population.[43] The study will take place in peri-urban, low-income areas in two regions in Peru (Manchay, Lima and Huánuco city, Huánuco district). In each site, we will work with one principal health centre and one subsidiary health centre (four in total) and the mothers and IYC in the surrounding population. Workshops will be held either in the health centre buildings or venues close to the health centres.

### Study participants

The participatory design process will include caregivers from the two peri-urban communities in Lima and Huánuco, the healthcare professionals of the participating health centres and the community health agents. Community health agents are members of the community who conduct home visits to mothers with IYC to follow-up and promote iron supplementation (including haemoglobin tests or sometimes delivering iron supplements to the home), vaccination uptake and attendance at health checks. Throughout this paper, we use 'community participants' to refer to caregivers, healthcare professionals and community health agents. Co-design workshops will include caregivers of IYC, staff of the community health centres with responsibility for nutritional counselling and growth and development checks and the community health agents. We will engage with community participants to explore, experiment, co-design, co-create and iteratively validate new design ideas to address the four major challenges around IYC nutrition identified in phase I. The community-based co-design workshops will take place in parallel recognising that the socioeconomic and cultural context as well as digital literacy in each study location might differ and influence differently the outcomes of the co-design process.[44 45]

### Sampling

We will invite 10–12 mothers or caregivers to each workshop in each site of Lima and Huánuco (ie, 20–24 participants in total for each stage of the co-design process)

through purposive sampling of mothers/caregivers registered at the health centres with IYC. Based on previous experience, we anticipate that not all invited participants will be able to attend on the day and have allowed for up to 50% non-attendance. If there is attrition or drop out between one workshop and the next, new participants from the same community will be invited to take part.

We will purposefully invite two healthcare professionals (including community health agents) to participate from each of the four health centres that are participating in Lima and Huánuco, aiming for at least three healthcare staff to attend each co-design workshop. Health centre participants will be recruited to represent the range of personnel involved in IYC healthcare namely, nutritionists, nurses responsible for growth and development checks and community health agents. Recruitment of healthcare professionals will be coordinated with the head of the health centres according to their time availability. Caregivers will be recruited by the community health agents who will invite mothers from the list of families assigned to them and send reminders by telephone and WhatsApp. An estimated 95% of households in Peru have access to a mobile phone.[46]

### Sample size calculation

Community-based co-design workshops employ primarily design and qualitative methods. There is no sample size calculation based on a quantitative outcome or measures related to user and design research methods. We aim to recruit a diversity of characteristics in the health professionals (men and women, different job roles and at different career stages/length of service in the area). Among caregivers, we aim to recruit participants with a range of characteristics relating to age, employment status, education and number of children. We will ensure accountability and rigour through continuous 'debate, critique and reflection'[47] between the participants and the research team.

### Data collection methods

Prior to the participatory co-design process, the research team will create a set of design artefacts/probes and drawings to actively support the engagement of the participants, such as a visual storyboard with different scenarios (scenario-based challenge cards) reflecting the top challenges from phase I, and a collection of ideation/design cards that includes a deck of pictorial representations of people, resources and strategies as well as technology from the phase I findings, research team suggestions and informed by relevant literature.

We will then conduct a series of four workshops in each site (eight in total), each one of approximately 2 hours duration on average including a refreshment break. The workshops will take place in parallel in the two study areas led by the same researchers to strengthen methodological consistency and continuity while also accounting for the contextual differences from each study setting to adapt methods when needed. Different design ideas

and co-produced artefacts can be developed in the two locations, but emerging design concepts and ideas will be cross-referenced across the co-design workshops in each site to allow for complementarity. Workshops will be audio-recorded and visual activities will be recorded by photography and workshop co-created materials. The sequence and purpose of workshops is outlined below.

### Idea generation workshops

Community participants (mothers, family members, health facility staff and community health agents) will generate creative and innovative ideas to support healthy diets for IYC including Double Duty Actions (actions to address micronutrient deficiency and anaemia, avoiding excess intake)[8] and will discuss how to implement recommended practices.[48] Methods will include using printed sketches with different scenarios and reflecting on the main challenges to healthy dietary practices identified in phase I. Scenarios will be read to participants who will then brainstorm using post-it notes and sketch potential solutions (supported by ideation/design cards) or interventions to assist family caregivers, government healthcare centres and staff to enhance nutrition and complementary feeding. Interventions could be digital/non-digital, targeting relevant users (ie, households, individuals or healthcare professionals) or the wider community or health service environment. At least three design themes will be drawn from the idea generation workshops following collective analysis and evaluation. These will be refined further using visual sketching techniques for conceptual design[49–51] and to enable future thinking.[52 53] A deciding vote will select the preferred design concepts to be considered in the next workshops.

### Co-creating future scenarios workshops

Taking a future-oriented perspective,[52] community participants will explore and co-create visions of the future/future scenarios through sketching techniques[53] and the 'futures' workshop method.[54 55] First, participants will critically reflect on the design ideas/concepts/sketches that emerged from the idea generation workshops. Next, in the 'fantasy' phase, participants will divide into small groups to devise new ideas and visions of the future for further development. Use of sketching will promote collaborative reflection, innovation and co-design of the ideal future.[56] Finally, participants will discuss the envisaged concepts/scenarios and assign implementation priorities based on suitability to the local communities and select up to three design concepts for further exploration as potential interventions.

### Storyboarding workshops

Participants will create storyboards[48] to further develop and visually convey up to three proposed design concepts. They will then elaborate and consolidate the best-case scenario for each concept. After presenting storyboards, participants will discuss the pros and the cons of each

concept and a list of key functionalities that stand out for each design concept will be identified.

### Early implementation and feedback workshops

Early stage implementation of the design concepts will take place through design plans, visualisations and creating material artefacts, design probes and intervention prototypes. Prototypes are proposals for future 'things-that-are-not-quite-objects-yet',[57] providing insights into what might be possible in the future to current configurations of resources, people and practices. These workshops will gather early feedback on the initial design iterations through, for example, feedback sessions and hands-on activities. At the end, up to one design concept per research site will be selected for further development based on early feedback. High-fidelity prototypes will be implemented and refined after the final workshop and before piloting.

### Pilot studies to refine designed intervention prototypes

We will pilot the materials, messages and delivery methods of the intervention prototype in the health facilities or households in the study areas. This may include new approaches to staff training, novel methods for delivering messages for both health service staff or parents/caregivers and/or in the home (eg, text messaging, games, etc). Prototypes will be piloted for 1–2 months. Qualitative evaluation will use ethnographic methods, data logging, prequestionnaires/postquestionnaires and interviews and will be adapted to the emerging characteristics of each prototype. The evaluations with community participants will explore relevance, acceptability, feasibility and impact potential of the refined version of the generated prototypes. Exit interviews with mothers attending health facilities will be used to evaluate their experiences. This project timescale is insufficient to evaluate dietary behaviour changes for IYC feeding, but subsequent funding will be sought to conduct further long-term evaluations and redesign and refinement of the prototypes.

### Data analysis

Audio-recordings of participatory workshops will be transcribed verbatim in Spanish by the same researcher who facilitates workshops. Observations, written notes and photographs from workshops will be recorded as further supporting materials. Transcripts of workshops will be analysed along with the materials produced, using a collaborative tool (Miro). At each workshop, we will share the summaries and themes with participants that have come from the previous workshop, using drawings and designs to capture ideas and themes, to support discussion, reflection and additional feedback, for the process of member-checking, or respondent validation. In-depth analysis of data will take place alongside the participatory design process. Coding will be conducted in Spanish primarily by one investigator (DO-R) who will then check and discuss codes with co-investigators (one expert in HCI (NV) and one nutritionist (RGP)) to reach consensus. Codes will then be translated to English by one of the bilingual researchers. Qualitative data will be analysed using a combination of affinity diagramming to quickly analyse the co-produced materials to identify requirements and complemented by thematic analysis of participants' discussions using NVivo software.

### Ethics and dissemination

Ethical approval for this study was obtained from the Ethical Review Committee of the Instituto de Investigación Nutricional (IIN) Peru (reference 388-2019/CIEI-IIN), Loughborough University (C19-87) and confirmed by Cardiff University based on local approval in Peru. Written informed consent, and reconsent for repeat workshop participation, will be provided by all participants after receiving written and verbal information about the study. Participants will be informed of the right to withdraw from participation at any stage.

Data (audio-recordings, transcripts, drawings, co-created materials, prototype designs and images) will be stored on a secure workspace at Loughborough University accessible only by investigators and designated research staff during the project. During data collection, information will be stored on encrypted or password-protected laptops and tablets in Peru. No personal identifying information will be associated with design outputs, and all drawings, ideas and creative output will be stored anonymously. Following the study, data will be stored on the institutional research repository at Loughborough University and made open access after a period of embargo. The tools used for the survey, in-depth interviews and policy mapping have been deposited on the research repository.[58–61] Data relating to phase I of the project can be accessed at https://doi.org/10.17028/rd.lboro.c.6329105.v1; https://doi.org/10.17028/rd.lboro.c.6329093.v1 and https://doi.org/10.17028/rd.lboro.c.6349121.v1 Publications will follow criteria for reporting qualitative research (Consolidated criteria for Reporting Qualitative research) guidelines[62] and Guidelines for Reporting the Impact of Patient and Public involvement in research (GRIPP2).[63]

Findings of the participatory design process will be disseminated through a deliberative workshop in Lima, Peru. Community participants and policy makers will be invited to the deliberative workshop with study investigators to consider the outcomes of the pilot work. This workshop will share insights from the iterative design process and the developed prototypes. We will reflect on what worked and what did not work during the design activities, identifying factors that facilitated success (eg, design probes (storyboard and design cards) showing great potential to facilitate community participation and ideation, active community engagement and continued participation, generation of creative and context-relevant ideas, future scenarios and design concepts, community participants valuing and trusting the co-design process, etc).[44 64] Research findings from phase I and phase II will

also be disseminated through academic publications. We will seek future opportunities to continue developing the generated prototypes and rolling out the prototypes as a trial intervention with long-term evaluation to examine the effect on complementary feeding behaviours and dietary outcomes for infant and young child nutrition.

## DISCUSSION AND CONCLUSIONS

The first phase of this study identified nutritional challenges relating to stunting, risk of overweight or obesity and anaemia in IYC among low-income, peri-urban communities in Peru. From these identified challenges, we prioritised the top four nutritional challenges to take forward to our participatory co-design study (phase II). This next phase will involve healthcare professionals, community health agents and caregivers working together to iteratively co-design and test prototypes that aim to address the identified nutritional challenges. The study will offer recommendations for design, research and policy in relation to the co-design of digital health tools in the context of IYC nutrition in Peru. An important consideration in the study is that our formative research was conducted up to March 2020 before the COVID-19 pandemic, and our participatory co-design phase started in June 2022 when most restrictions relating to COVID-19 were lifted. The intervening period may have influenced caregiver and healthcare professionals' ideas and perceived needs for healthy nutrition of IYC. Working across two languages, English and Spanish, also adds some complexities to the co-design process. We also recognise that the participatory co-design work conducted in Lima and Huánuco may not capture the nutritional challenges for all urban contexts in Peru, and future work would need to examine how our co-design approach and emerging prototypes could be adapted for other urban and rural environments.

**Acknowledgements** We gratefully acknowledge the contribution of all the investigators and field and research assistants on the PERUSANO project. We also thank the community participants, stakeholders and participating health centres.

**Contributors** ER, HMC-K, NV, RGP, RB, MH and PG designed and led the development of the funded grant proposal. ER, RGP, HMC-K, NV, RB, MH, PG, RP and DO-R developed detailed protocols for phase I and the participatory co-design (phase II). All authors contributed to writing the manuscript and had critical input on the final version.

**Funding** This study was supported by the UK Medical Research Council (MR/S024921/1) and CONCYTEC/ PROCIENCIA Perú (032-2019-FONDECYT) through the Newton Fund.

**Competing interests** None declared.

**Patient and public involvement** Patients and/or the public were involved in the design, or conduct, or reporting, or dissemination plans of this research. Refer to the 'Methods' section for further details.

**Patient consent for publication** Not applicable.

**Ethics approval** Ethical approval for this study was obtained from the Ethical Review Committee of the Instituto de Investigación Nutricional (IIN) Peru (388-2019/CIEI-IIN), Loughborough University (C19-87) and confirmed by Cardiff University. Participants gave informed consent to participate in the study before taking part.

**Provenance and peer review** Not commissioned; externally peer reviewed.

**Data availability statement** Data are available in a public, open access repository.

**ORCID iD**
Emily Rousham http://orcid.org/0000-0001-5654-9279

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
