## [Reviewer comments · BMJ Open]

ARTICLE DETAILS

TITLE (PROVISIONAL)	Designing intervention prototypes to improve infant and young child nutrition in Peru: A participatory design study protocol
AUTHORS	Rousham, Emily; Pareja, Rossina G.; Creed-Kanashiro, H.M.; Bartolini, R.; Pradeilles, Rebecca; Ortega-Roman, Deysi; Holdsworth, Michelle; Griffiths, Paula; Verdezoto, Nervo

VERSION 1 – REVIEW

REVIEWER	Genevieve Stone Global Alliance for Improved Nutrition
REVIEW RETURNED	23-Apr-2023

GENERAL COMMENTS	Page 14, line 26: the authors discuss that they will identify factors that 'facilitated success.' It's not clear 'what' is considered as 'success' from the activities or how the 'success' of the workshops is being measured. There are also a couple of suggested minor amendments to text: Page 6, line 5: Remove 'b) and c)' from this part of the sentence, as they are included later in the sentence and they're a bit confusing when used at line 5. Page 10, line 42: needs the word 'to' in this part of the sentence: '...characteristics relating to age,...'
---

REVIEWER	Jorge Silva Parties Fiestas Universidad Católica Santo Toribio de Mogrovejo, Lambayeque
REVIEW RETURNED	05-Jun-2023

GENERAL COMMENTS	Necesita aclarar limitaciones de la población ya que están trabajando con dos poblaciones que a pesar que pueden tener características similares, son dos departamentos que difieren en el aspecto de cultura, economía, creencias, nivel académico, etc. Por lo tanto los resultados pueden variar entre estas dos poblaciones por estas variables que el investigador no puede manejar, pero si aclararlas.
---

VERSION 1 – AUTHOR RESPONSE

Reviewer: 1

Comment: Page 14, line 26: the authors discuss that they will identify factors that 'facilitated success.' It's not clear 'what' is considered as 'success' from the activities or how the 'success' of the workshops is being measured.

Response: thank you for highlighting this, we have expanded on what factors will be considered as indicators of success or otherwise. We do not have a quantitative measure of success for the workshops, but will undertake a qualitative evaluation of the prototypes co-designed through the

workshops in terms of their acceptability and usability as well as reflections on how the co-design process enabled participation and innovation to community participants. See page 14, lines 13-17.

(e.g., design probes (storyboard and design cards) showing great potential to facilitate community participation and ideation, active community engagement and continued participation, generation of creative and context-relevant ideas, future scenarios, and design concepts, community participants valuing and trusting the co-design process, etc) (43,59).

Comment-There are also a couple of suggested minor amendments to text: Page 6, line 5: Remove 'b) and c)' from this part of the sentence, as they are included later in the sentence and they're a bit confusing when used at line 5.

Response: Thank you, we have deleted the duplicated letters 'b) and c)' and made the sentence clearer-see Page 6, Lines 1 & 3.

Comment-Page 10, line 42: needs the word 'to' in this part of the sentence: '...characteristics relating to age,...'

Response: This has been corrected.

Reviewer: 2

Comment: 'Necesita aclarar limitaciones de la población ya que están trabajando con dos poblaciones que a pesar que pueden tener características similares, son dos departamentos que difieren en el aspecto de cultura, economía, creencias, nivel académico, etc. Por lo tanto los resultados pueden variar entre estas dos poblaciones por estas variables que el investigador no puede manejar, pero si aclararlas.'

We have translated this to English as 'You need to clarify population limitations since you are working with two populations that, although they may have similar characteristics, are two departments that differ in terms of culture, economy, beliefs, academic level, etc. Therefore, the results may vary between these two populations due to these variables that the researcher cannot manage, but can clarify.'

Response: Thank you and we agree with the reviewer. We stated in the original manuscript that *'the workshops will take place in parallel in the two study areas led by the same researchers to strengthen methodological consistency and continuity. Different design ideas can be developed in the two locations, but emerging design concepts and ideas will be cross-referenced across the co-design workshops in each site to allow for complementarity'*, but we have now added *'while also accounting for the contextual differences from each study setting to adapt methods when needed'*. see Page 10-11:

We have also expanded on the study design to explain that the workshops will be conducted in parallel in each site whilst allowing local influences and contexts to vary, Page 9, paragraph 2, lines 14-17:

'The community-based co-design workshops will take place in parallel recognising that the socioeconomic and cultural context as well as digital literacy in each study location might differ and influence differently the outcomes of the co-design process (43,44).'

And also Page 12, para 3, lines 6-7 *'and will be adapted to the emerging characteristics of each prototype'*